# Intravenous Infection of Small Ruminants Suggests a Goat-Restricted Host Tropism and Weak Humoral Immune Response for an Atypical Bluetongue Virus Isolate

**DOI:** 10.3390/v15010257

**Published:** 2023-01-16

**Authors:** Massimo Spedicato, Giovanni Di Teodoro, Liana Teodori, Mariangela Iorio, Alessandra Leone, Barbara Bonfini, Lilia Testa, Maura Pisciella, Claudia Casaccia, Ottavio Portanti, Emanuela Rossi, Tiziana Di Febo, Nicola Ferri, Giovanni Savini, Alessio Lorusso

**Affiliations:** Istituto Zooprofilattico Sperimentale dell’Abruzzo e del Molise, 64100 Teramo, Italy

**Keywords:** bluetongue, atypical serotype, goat, sheep

## Abstract

Bluetongue virus (BTV) is the etiologic agent of bluetongue (BT), a viral WOAH-listed disease affecting wild and domestic ruminants, primarily sheep. The outermost capsid protein VP2, encoded by S2, is the virion’s most variable protein, and the ability of reference sera to neutralize an isolate has so far dictated the differentiation of 24 classical BTV serotypes. Since 2008, additional novel BTV serotypes, often referred to as “atypical” BTVs, have been documented and, currently, the full list includes 36 putative serotypes. In March 2015, a novel atypical BTV strain was detected in the blood of asymptomatic goats in Sardinia (Italy) and named BTV-X ITL2015. The strain re-emerged in the same region in 2021 (BTV-X ITL2021). In this study, we investigated the pathogenicity and kinetics of infection of BTV-X ITL2021 following subcutaneous and intravenous infection of small ruminants. We demonstrated that, in our experimental settings, BTV-X ITL2021 induced a long-lasting viraemia only when administered by the intravenous route in goats, though the animals remained healthy and, apparently, did not develop a neutralizing immune response. Sheep were shown to be refractory to the infection by either route. Our findings suggest a restricted host tropism of BTV-X and point out goats as reservoirs for this virus in the field.

## 1. Introduction

Bluetongue virus (BTV) (genus *Orbivirus*, family *Sedoreoviridae*) is the etiologic agent of bluetongue (BT), a World Organization for Animal Health (WOAH, funded as OIE)-listed infectious disease affecting wild and domestic ruminants, primarily sheep (*Ovis aries*) [1]. With very limited exceptions [2,3,4], BTV is almost exclusively transmitted among its natural hosts by female biting midges of the genus *Culicoides*, and thus outbreaks mostly occur in late summer to early autumn [5,6].

The BTV genome is encased in a triple-layered capsid and consists of 10 dsRNA segments (from S1 to S10) coding for 7 structural proteins (VP1-7) and 5 non-structural proteins (NS1, NS2, NS3/NS3a, NS4, and S10-ORF2) [7,8,9]. 

The classification of BTV into serotypes is largely dictated by the outermost and highly variable capsid protein VP2, encoded by S2, and by the ability of reference sera to neutralize an isolate. In this way, twenty-four historical serotypes have been recognized worldwide. BT has been reported multiple times in the last twenty-four years in the European Union (EU), causing devastating BT outbreaks in sheep and cattle and huge economic losses due to diseased animals and restrictions on animal movement and trade. Most BT outbreaks had a direct northern African origin, most likely due to wind-driven dissemination of infected midges from northern African countries, where multiple BTV serotypes are endemic [10,11,12,13,14]. 

Since 2008, largely due to the availability of next-generation sequencing technology, 12 additional putative serotypes have been detected and partially characterized. [15]. Some of these novel serotypes (from BTV-25 onward) could not be typed by classical neutralization methods (lack of reference sera, inability of the virus to grow on cells), and assignment to a specific serotype relied on S2 sequence analysis alone [16].

The novel serotypes have so far only been detected in small ruminants, although antibodies neutralizing BTV-26 have been detected in cattle and dromedaries from Mauritania [12]. They are often referred to as “atypical” due to their unusual in vivo and in vitro characteristics compared to those of classical serotypes, including a null or minimal virulence (they indeed do not cause the bluetongue disease), the inability of some of them to grow on cell cultures, the possibility of vector-free transmission, and long-lasting infection in small ruminants [14,15,16,17,18,19,20,21,22,23,24]. 

In March 2015, during BTV surveillance activities in Italy, a novel uncultivable atypical BTV strain was detected in the blood of asymptomatic sentinel goats located in two unrelated goat flocks in Sardinia [21] and named BTV-X ITL2015 (putative serotype 32; [23]). A novel strain named BTV-X ITL2021, likely belonging to the same serotype due its high genetic relatedness to BTV-X ITL2015, was detected in Sardinia in 2021, but, unlike BTV-X ITL2015, was isolated in cell culture [25]. 

Therefore, to further characterize the role of BTV-X ITL2021 in small ruminants, we conducted two animal experiments in sheep and goats to investigate viral pathogenicity, kinetics of infection, and host humoral immune response. 

## 2. Materials and Methods

### 2.1. Ethics

All animal experiments were performed according to Directive 2010/63/EU and approved by the Italian Ministry of Health (authorization No. 76/2019-PR). Before infection, animals were allowed a 5-day acclimatization period. 

### 2.2. Animals and Virus

Sheep and goats (*Capra aegagrus hircus*) (adult females, mixed breeds) were housed (each species in separated pens) in the insect-proof BSL-3 facilities at the Istituto Zooprofilattico Sperimentale dell’Abruzzo e del Molise (IZS-Te). All animals were purchased from local vendors and determined to be healthy and free from BTV RNA and antibodies by real-time RT-PCR (Hofmann et al., 2008 (RT-qPCR_NS3_; [17]) and competitive ELISA (c-ELISA), respectively.

The BTV-X ITL2021 strain (herein referred to as BTV-X; accession numbers NCBI MZ325474-MZ325483) was isolated in BSR cells from a blood sample taken from a goat in Italy in 2021 [25]. Viral passages n. 3 and n. 4 were used for infection in Trial 1 and Trial 2, respectively. Control animals were sham-infected and maintained throughout the study. 

### 2.3. Infection and Sampling

In Trial 1, 5 sheep (S11 to S15) and 5 goats (G11 to G15) were inoculated subcutaneously (SC), at the inner face of the thigh, at day post-infection (dpi) 0 with 1 mL of Minimum Essential Medium (MEM; Sigma Aldrich, St. Louis, MO, USA) containing 10^6.13^ tissue culture infectious doses 50 (TCID_50_)/mL of BTV-X. Three individuals (S21-23, G21-23) for each species were left uninfected in the same pen and served as direct-contact animals. In addition, one individual (S24 and G24) of each species was housed in a different pen and used as an indirect control animal. Rectal temperature and clinical signs were checked daily for two weeks. A temperature ≥40 °C was considered as a febrile state.

Serum and EDTA-blood samples, rectal, nasal, and oral swabs (LP Italiana SPA, Milan, Italy), were collected from all animals 3 times a week from dpi0 to dpi59. Swabs were soaked in 2 mL of MEM immediately after sampling. 

At dpi59, based on the results obtained from analysis of all biological samples, Trial 1 was discontinued, and a second trial (Trial 2) was scheduled to test the intravenous route (IV) of infection, as follows. Three sheep (S21 to S23) and three goats (G21 to G23), used as direct contact animals in Trial 1 and demonstrated to be negative for BTV-RNA and antibodies, were inoculated IV (jugular vein) at dpi0 with 1 mL of MEM containing 10^6.29^ TCID_50_ of BTV-X.

Back titration was performed, and the results confirm the inoculated titre for both trials, with very minimal and irrelevant deviations of the values, due to the normal variability of the test. Rectal temperature and clinical signs were checked daily for two weeks, where a temperature ≥40 was considered as fever. Sera and EDTA-blood samples were collected from all animals 3 times a week from dpi0 to dpi39 and once a week from dpi44 to dpi135. As soon as BTV RNA was detected in the blood, the collection of oral, nasal, and rectal swabs was implemented when blood sample were taken.

### 2.4. Analyses on EDTA-Blood Samples and Swabs

Total RNA was extracted (QIAamp Viral RNA Kit; Qiagen, Hilden, Germany) from EDTA blood samples and swabs’ transport medium (after manual shaking) and tested for BTV-X RNA with a modified in-house real-time RT-PCR (RT-qPCR_BTV-X_; [21]). 

### 2.5. Virus Isolation and Titration

All RT-qPCR_BTV-X_-positive blood samples were propagated in BSR cells (a clone of BHK), as described in [15]. A total of 500 microliters of EDTA blood was suspended in phosphate-buffered saline (PBS) and centrifuged at 800 rpm for 2 min. Red blood cells were then washed twice in PBS, resuspended in PBS, and sonicated for approximately 20 s to allow cell lysis and virus release. Lysed blood (0.2 mL) was then inoculated on a pre-seeded BSR cells monolayer and incubated for 2 h at 37 °C. The inoculum was then replaced with MEM containing 10% foetal calf serum. After 5 days of incubation, the supernatant of the infected cell flask was collected separately, whereas the BSR cell monolayer was treated with trypsin to allow detachment.

Once detached, the cells were suspended with 5 mL of the retained supernatant and the mixture was transferred to a new pre-seeded BSR cell monolayer for 5–7 days. Three passages on BSR cells were performed. After CPE development, cells were scraped off and the success of viral isolation was confirmed by RT-qPCR_BTV-X_.

End-point virus titration of BSR-isolated strains was performed using BSR cells [26], starting from 1 log10 dilution of the blood. The TCID_50_ was calculated using the method of Reed and Munch [27].

### 2.6. Analyses on Serum Samples

Serum samples were screened for anti-BTV VP7 antibodies with an in-house competitive ELISA (cELISA; see Table 1 for the details of the assay), and neutralizing antibodies were titrated by virus neutralization (VN; WOAH *Manual of Diagnostic Tests and Vaccines for Terrestrial Animals 2019;* https://www.woah.org/fileadmin/Home/eng/Health_standards/tahm/3.01.03_BLUETONGUE.pdf; accessed on 13 December 2021) against the BTV-X strain (starting with a serum dilution of 1:10). Control serum samples were used in the assay by enrolling BTV-X-positive serum samples of goats. The neutralization titer was defined as the reciprocal of the highest dilution without any CPE in the wells.

## 3. Results

### 3.1. Subcutaneous Infection Did Not Elicit a Productive Infection

Neither fever (T ≥ 40 °C) nor clinical signs were observed in animals throughout the trial. BTV RNA was not detected by RT-qPCR_BTV-X_ in all animals regardless of the type of biological specimen. Serum samples of sheep and goats were also negative by c-ELISA throughout the sampling period. Therefore, based on these combined results, Trial 1 was discontinued at dpi59. Direct- and indirect-contact animals were also negative and therefore enrolled for Trial 2. 

### 3.2. Intravenous Infection Elicited a Productive Infection Only in Goats

All animals remained healthy and afebrile throughout the experimental period. BTV-X RNA was not detected in all specimens from sheep and only sporadic detection of antibodies by c-ELISA was revealed in ovine serum samples. Neutralizing antibodies were not quantifiable by VN. Accordingly, sampling from sheep was discontinued at dpi44. 

Conversely, BTV-X RNA was first detected in all goats at dpi11 (Ct 28–31; mean Ct 29.3) and peaked (lowest Ct) between dpi11 and dpi14 (mean peak, Ct 28.3 at dpi14). All three goats remained positive for BTV-X RNA till the end of the trial (dpi135; Ct 36–37; mean 36.7), though a gradual decreasing trend was observed. The virus was isolated from blood samples in BSR cells, starting from dpi11 in G22 and G23 and from dpi14 in goat G21. The virus was isolated up to dpi30 in G23 (Table 2). Titration of the virus straight from the blood was always unsuccessful (limit of detection of the method was < 1.3 log10 TCID_50_/mL). BTV-RNA was not detected in nasal, rectal, or oral swabs at any time point. 

Goat G23 died 2 days after the end of the trial (corresponding to dpi137) for reasons apparently not related to the trial. Nevertheless, spleen, lungs, kidneys, and mediastinal lymph nodes tissues were tested and found negative for the presence of BTV-X RNA. 

All goats seroconverted by dpi16, five days after the first detection of BTV-X RNA, as determined by the cELISA, and remained cELISA-positive for the entire sampling period. No animals showed detectable neutralizing antibodies as measured by VN (Table 3).

## 4. Discussion

In this study, we inoculated, by two different routes, small ruminants to investigate the pathogenicity, the kinetics of infection, and the host humoral immune response of an atypical BTV serotype originally isolated from goats in Sardinia, Italy. In the first trial, BTV-X, inoculated by the SC route into goats and sheep, did not elicit a productive infection as BTV-X RNA was not detected in blood or swab samples at any time point in either species. Furthermore, none of the animals seroconverted. The scenario radically changed when goats were inoculated by the IV route. Indeed, in this species, BTV-X RNA was detected in blood samples starting at dpi11 and remained constant till the end of the sampling period, although with a decreasing trend. 

In contrast to the classical serotypes, where viral RNA can be detected as early as 2–3 days post-infection [4,28,29,30,31], BTV-X RNAemia started later but with a higher Ct, in accordance with results obtained for goats experimentally infected with BTV-27 (with which BTV-X shares the highest VP2 sequence identity) [21], wherein virus-specific RNAemia was demonstrated to start at 7 to 14 days post-infection [32]. Conversely, in our study, sheep remained RT-qPCR_BTV-X_-negative and did not seroconvert to BTV-X. This finding is in accordance with those reported after infection of sheep with a chimeric BTV-25 [33] and with BTV-27 [32], both of which were originally identified in goats. 

The different outcomes observed in sheep and goats are suggestive of a restricted host tropism of BTV-X for goats, though definitive conclusions cannot be drawn. In this regard, cattle were not included in the trials, and a recent serological survey conducted with samples collected from domestic ruminants in Sardinia suggested the presence of antibodies neutralizing BTV-X in vitro in cattle and small ruminant serum samples [25]. 

RNAemia was long-lasting (up to dpi135), but as expected, isolation of infectious virus was achieved only for a shorter period (up to dpi 30 at the most). It is well known that only infectious BTV, as determined by virus isolation, is required for *Culicoides* infection and onward transmission to naïve animals, but whether infectious viraemia is involved in BTV-X transmission is currently unknown. In our experimental settings, nasal, rectal, and oral swabs were negative for BTV-X; thus, a contact transmission, as demonstrated for BTV 26 [34] and BTV27v02 [32], remains speculative for BTV-X.

BTV-X’s kinetics were deeply influenced by the route of inoculation, such that BTV-X led to a productive infection only when inoculated by the IV route and not by the SC route, the latter being the most similar route to natural infection and indeed commonly used in experimental trials with classical BTV serotypes. It is important to point out that in a small-scale vector competence study, in which field-collected *Culicoides* of the Obsoletus complex were fed with BTV-X-infected blood, it was suggested that *Culicoides* midges are probably not competent vectors for BTV-X (Savini, personal communication), corroborating the evidence for the unfeasibility of propagating BTV-X on *Culicoides*-derived continuous cells [21,25]. 

The nature of the serological response in goats in this study contrasts with that observed in infections with extant atypical serotypes. Indeed, the virus-specific seroconversion detected by c-ELISA did not include a virus-neutralizing antibody response. This discrepancy is hard to explain. BTV-X could be poorly immunogenic, a scenario already described for BTV-25 and BTV-33 [23,35]. It was presumed the IV administration was able to circumvent the skin-to-local lymph node pathway and therefore the disruption of follicular dendritic cells hindering B-cell division in germinal centers, resulting in a delayed or reduced production of neutralizing antibodies [36,37,38]. This unexpected result highlights the minimal data that are currently available for atypical BTV serotypes and the need for further pathogenesis studies involving this group of atypical viruses. 

In conclusion, this work demonstrated that BTV-X, originally identified in goats, can experimentally induce infectious viraemia only in goats and not in sheep only when administered by the IV route. One important shortcoming of this study is the absence of direct and indirect contact animals in Trial 2, although it is critical to point out the absence of viral RNA in all swabs taken from viraemic animals at all time points. Goats remained healthy throughout the study period and had long-lasting RNAemia, and these findings corroborate the hypothesis that this species can act as a reservoir for this virus in the field. 

## Figures and Tables

**Table 1 viruses-15-00257-t001:** Details of the in-house competitive ELISA used in the study.

Competitor	Cut-off for Positivity/Negativity	Initial Serum Dilution	Sensitivity (C.I. 95%)	Specificity (C.I. 95%)
Anti-BTV VP-7 monoclonal antibody conjugated with horseradish peroxidase (Mab-HRP)	Results are expressed as percent inhibition using the following formula: (mean OD test serum/mean OD of the Mab) × 100. Sera with a percent inhibition value > 50% are considered negative; sera with a percent inhibition value ≤ 50% are considered positive.	Sera are not diluted	100% (95–100%)	100% (97.6–100%)

**Table 2 viruses-15-00257-t002:** Real-time RT-PCR threshold cycles (Ct) in the blood samples of the three goats intravenously infected with BTV-X (Trial 2). Negative samples are shown as 45. Bold red numbers represent the peak Ct for the goat and the group. Shaded boxes indicate samples from which virus was isolated on cell culture.

	Days Post-Infection																													
Goat id	0	4	7	11	14	16	18	21	23	25	28	30	32	35	37	39	44	51	58	65	72	79	86	93	100	107	114	121	128	135
G21	45	45	45	31	**29**	**29**	**31**	31	30	32	32	31	32	30	29	30	29	36	32	32	34	34	33	34	35	34	32	34	33	36
G22	45	45	45	**29**	**28**	**28**	**31**	32	30	32	**31**	31	33	31	30	30	34	35	33	34	35	33	33	36	34	35	33	36	33	37
G23	45	45	45	**28**	**28**	**29**	**33**	31	**33**	31	31	**32**	32	33	32	31	33	35	34	35	34	34	34	35	38	36	33	38	34	37
Mean Ct	45.0	45.0	45.0	29.3	**28.3**	28.7	31.7	31.3	31.0	31.7	31.3	31.3	32.3	31.3	30.3	30.3	32.0	35.3	33.0	33.7	34.3	33.7	33.3	35.0	35.7	35.0	32.7	36.0	33.3	36.7

**Table 3 viruses-15-00257-t003:** cELISA results of the three goats intravenously infected with BTV-X (Trial 2). Day post-infection 11 was the first day BTV-X RNA was detected in goats. No animals showed detectable neutralizing antibodies as measured by VN.

	Days Post-Infection																													
Goat id	0	4	7	11	14	16	18	21	23	25	28	30	32	35	37	39	44	51	58	65	72	79	86	93	100	107	114	121	128	135
G21	neg	neg	neg	neg	neg	**pos**	**pos**	**pos**	**pos**	**pos**	**pos**	**pos**	**pos**	**pos**	**pos**	**pos**	**pos**	**pos**	**pos**	**pos**	**pos**	**pos**	**pos**	**pos**	**pos**	**pos**	**pos**	**pos**	**pos**	**pos**
G22	neg	neg	neg	neg	neg	**pos**	**pos**	**pos**	**pos**	**pos**	**pos**	**pos**	**pos**	**pos**	**pos**	**pos**	**pos**	**pos**	**pos**	**pos**	**pos**	**pos**	**pos**	**pos**	**pos**	**pos**	**pos**	**pos**	**pos**	**pos**
G23	neg	neg	neg	neg	neg	**pos**	**pos**	**pos**	**pos**	**pos**	**pos**	**pos**	**pos**	**pos**	**pos**	**pos**	**pos**	**pos**	**pos**	**pos**	**pos**	**pos**	**pos**	**pos**	**pos**	**pos**	**pos**	**pos**	**pos**	**pos**

## Data Availability

The data presented in this study are available in the article.

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
