# Peer review of "Intravenous Infection of Small Ruminants Suggests a Goat-Restricted Host Tropism and Weak Humoral Immune Response for an Atypical Bluetongue Virus Isolate"

_viruses, 2023, doi:10.3390/v15010257_

Round 1
Reviewer 1 Report
This manuscript describes a series of conventional animal inoculation experiments on small ruminants involving an Italian isolated representative (BTV-X ITL2021) of the more recently isolated group of 'atypical' bluetongue viruses. The authors intended to investigate virus pathogenicity and the nature of infection and the host immune response. They investigated two routes of virus delivery, subcutaneous and intravenous inoculation. Only the intravenous route elicited meaningful evidence of viral infection in goats but not sheep. This finding, based on the work of others, appears to be a common feature of many atypical BTVs. The lack of detection of a virus neutralizing response within the seroconversions seen in goats is intriguing and as pointed out by the authors, shows there is much to still investigate and understand about the nature of this group of viruses generally. Given this observation, continuing studies like the one undertaken here are needed to better understand the unique properties, pathogenic potential, host range and in particular modes of viral transmission within the atypical BTV group. The manuscript therefore warrants publication once the authors have attended to the grammatical corrections highlighted in the attached pdf document.

Author Response
Dear Reviewer,
we sincerely thank you for the time spent for reviewing our manuscript. We revised the manuscript according to your suggestions and uploaded the revised manuscript for your reference. All changes in the revised manuscript were tracked

Reviewer 2 Report
This article investigate the pathogenicity and kinetics of experimental infection (subcutaneous and intravenous routes) of sheep and goat with an atypical BT virus isolated in Italia. Sheep were shown refractory to the infection and this study suggest a restricted host tropism of BTV-X.
The topic is fascinating, and there are still many questions regarding atypical serotypes of Bluetongue.
However, after reading the article, especially in regards to the disease's transmission, more questions remain than there are answers.
I believe that this situation could be explained by the fact that the subcutaneous infection is inconclusive and there isn't any proof of horizontal transmission.
By attempting to provide hypotheses and perhaps suggesting additional studies, it would be interesting to further develop the discussion section.
Given that the isolation ability of the BTV-X-Ita-2015 and 2021 strains differ, I advise against extrapolating the results of this experiment to all BTV-X.
Author Response
We really appreciate the comments of the Reviewere. We uploaded a revised version of the manuscript, with the changes tracked.
